# API: Adaptive Prototype Imputation for Incomplete Multimodal Sentiment Analysis

Xiaotao Wang [* 1]   Yiyang Fang [* 1]   Wenke Huang [1]   Bin Yang [1]   Guancheng Wan [1]   Mang Ye [1]

## Abstract

Multimodal sentiment analysis aims to infer human emotions by integrating signals from diverse modalities. However, missing modalities are common in real-world applications due to sensor failure, data corruption, or privacy concerns. Existing approaches typically follow two main paradigms: recovery-based and non-recovery-based methods. This dichotomy results in two critical limitations: I) computational inefficiency and semantic inconsistency (recovery-based methods rely on heavy generators that incur prohibitive inference latency and risk semantic drift due to lack of class-level priors); II) lack of instance specificity (non-recovery-based methods rely on static global mappings that fail to capture sample-specific affective cues). To address these gaps, we propose **A**daptive **P**rototype **I**mputation (API). To mitigate I), we introduce *Semantic-anchored Class-Temporal Prototype Estimation (SCOPE)* to construct non-trainable prototypes as stable semantic anchors, promoting semantic reliability. To resolve II), we design *Directional Instance-Adaptive Affine Modulation (DIAM)* to dynamically modulate these anchors via direction-specific affine transformations, capturing instance-unique affective characteristics without generative overhead. Experimental results on CMU-MOSI and CMU-MOSEI demonstrate that API outperforms state-of-the-art baselines, establishing a robust and lightweight prototype-centric paradigm for multimodal sentiment analysis. The code is publicly available at https://github.com/KX-yolo/API.

*Equal contribution [1]National Engineering Research Center for Multimedia Software, School of Computer Science, Wuhan University. Correspondence to: Mang Ye <yemang@whu.edu.cn>.

*Proceedings of the 43rd International Conference on Machine Learning*, Seoul, South Korea. PMLR 306, 2026. Copyright 2026 by the author(s).

## 1. Introduction

Multimodal sentiment analysis (MSA) aims to infer human affective states and opinions from heterogeneous signals such as language, vision, and acoustics (Baltrušaitis et al., 2019; Das & Singh, 2023; Gandhi et al., 2023; Liang et al., 2024; Fang et al., 2025a;b). By jointly modeling verbal, vocal, and visual cues, MSA can capture affective evidence that may be ambiguous when observed from a single channel, enabling a more comprehensive understanding of human attitudes and behaviors. Driven by sophisticated fusion architectures (Li et al., 2023; Poria et al., 2018; Hazarika et al., 2020; Yang et al., 2022; Yu et al., 2021), MSA has enabled critical applications ranging from social media monitoring to real-time telehealth consultations in healthcare.

However, the transition from controlled laboratories to real-world deployments is frequently hindered by data incompleteness due to sensor failures, transmission errors, or privacy constraints. Crucially, this incompleteness does not merely reduce information quantity; it fundamentally disrupts the *cross-modal synergy* required for robust prediction, causing standard models to falter when expected semantic interactions are severed. This practical setting further requires incomplete MSA models to satisfy multiple constraints: preserving affective semantics, adapting to instance-specific cues, and maintaining efficient inference; together, these constraints motivate revisiting existing paradigms. Existing efforts have generally diverged into two paradigms: recovery-based and non-recovery-based methods, each addressing only part of the problem.

Recovery-based methods (Wang et al., 2023b; Zhang et al., 2025; Ma et al., 2021; Wang et al., 2023a) reconstruct the missing modality with a generator. Representative work such as IMDer (Wang et al., 2023b) employs a diffusion generator conditioned on observed modalities to reduce semantic ambiguity and improve distributional fidelity. As illustrated in Fig. 1(a), a typical recovery-based pipeline takes the available text and visual cues as conditions and attempts to reconstruct the missing audio feature through a diffusion model. However, in the absence of explicit *class-level statistical priors*, the recovered audio may deviate from the target sentiment region, drifting away from the intended positive class toward an inconsistent affective state.

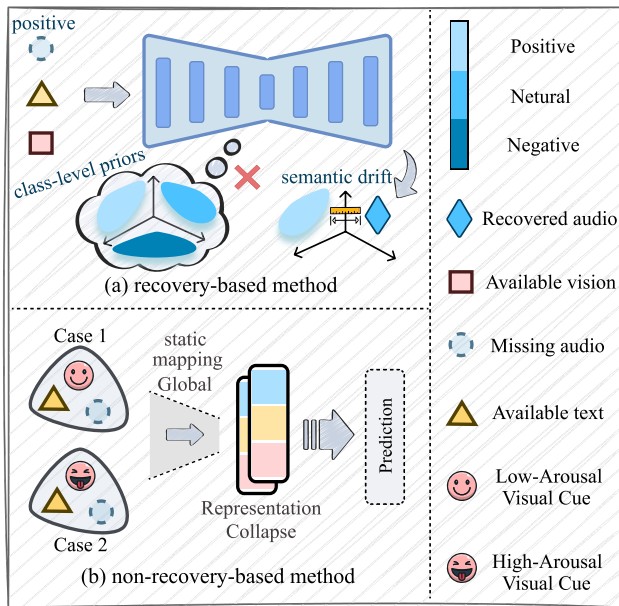

*Figure 1.* Limitations of existing paradigms for incomplete MSA. (a) Recovery-based methods incur significant inference latency from the multi-step denoising process and risk semantic drift due to lacking class-level priors. (b) Non-recovery methods employ a static, global projection, mapping distinct low- and high-arousal visual cues (Case 1 & 2) to indistinguishable features, causing representation collapse.

This semantic drift can undermine the cross-modal synergy required for reliable sentiment prediction. This mismatch is especially problematic for fine-grained sentiment regression, where small shifts in affective intensity can change the predicted score. Moreover, diffusion denoising introduces substantial computational costs, raising a pivotal problem: **I)** *How can we better preserve the semantic consistency of imputed features with the target affective class while meeting the low-latency constraints of real-time applications?*

To circumvent generation, non-recovery approaches (Lin & Hu, 2023; Pham et al., 2019; Li et al., 2024) learn a direct mapping from available observations to a unified latent representation. For instance, CorrKD (Li et al., 2024) distills correlation-decoupled knowledge to transfer structural semantics from complete to incomplete views, while Miss-Modal (Lin & Hu, 2023) employs geometric contrastive learning to align distributions across diverse missing conditions. However, these methods typically rely on a static projection shared across instances to handle missing patterns. Fig. 1(b) illustrates this limitation with two incomplete cases: both have similar available textual cues and missing audio, but their visual cues convey different affective intensities, namely low-arousal versus high-arousal states. A static global mapping may project these cases into overly similar latent representations, causing representation collapse and weakening instance-specific affective nuances. More generally, such static projections can over-

look instance-specific cross-modal correlations, producing generic representations that fail to capture subtle cues manifested in specific tones or micro-expressions. This raises a pivotal problem for real-time incomplete MSA deployments: **II)** *How can we dynamically capture instance-specific affective nuances without resorting to computationally intensive generative processes?*

Overall, both paradigms present a dilemma: representative diffusion-based recovery methods, such as IMDer, can reconstruct missing modalities but may suffer from semantic drift and substantial computational overhead, while non-recovery methods avoid iterative generation but may produce overly generic representations under static mappings. Despite these efforts, there remains a need for a unified framework that better balances semantic consistency, instance-specific expressiveness, and computational efficiency in practical incomplete MSA settings.

To bridge this gap, we propose a unified prototype-centric framework, **A**daptive **P**rototype **I**mputation (API), which reconceptualizes imputation not as a generative task but as an efficient *retrieval-and-modulation* process, decoupling semantic recovery from heavy computation. To address Problem I), we propose *Semantic-anchored Class-Temporal Prototype Estimation (SCOPE)*. SCOPE empirically accumulates training features to construct non-trainable, per-modality and per-class temporal prototypes. These serve as stable semantic anchors that guide imputed features toward valid semantic regions, reducing semantic drift without generative overhead. To address Problem II), we introduce *Directional Instance-Adaptive Affine Modulation (DIAM)*. By employing direction-specific lightweight MLPs (such as text-to-audio), DIAM generates affine parameters to modulate the retrieved prototypes. This direction-aware design helps the model capture sample-specific nuances and handle diverse missing patterns.

Our principal contributions are summarized as:

❶ To address Problem I), we propose Semantic-anchored Class-Temporal Prototype Estimation (SCOPE), which constructs non-trainable, per-modality and per-class temporal prototypes as semantic anchors, promoting semantic consistency without generative overhead.

❷ To address Problem II), we introduce Directional Instance-Adaptive Affine Modulation (DIAM), which dynamically calibrates retrieved prototypes through direction-specific transformations, capturing instance-specific affective nuances under diverse missing patterns for adaptive imputation.

❸ We conduct extensive experiments demonstrating that API consistently outperforms previous methods, achieving robust performance with low inference latency and negligible parameter overhead.

## 2. Related Work

### 2.1. Multimodal Sentiment Analysis

Multimodal Sentiment Analysis (MSA) aims to synthesize affective cues from language, vision, and acoustics to quantify human sentiment. While deep learning has advanced multimodal fusion (LeCun et al., 2015; Ye et al., 2022; Huang et al., 2023; Bai et al., 2025), most frameworks (Morency et al., 2011; Zadeh et al., 2017; Tsai et al., 2019; Chen et al., 2023a; Zhang et al., 2023; Zhu et al., 2023; Sun et al., 2024; Yang et al., 2024; 2025; Fang et al., 2026) assume modality completeness during training and inference. However, their performance precipitates substantially when modalities are incomplete, a pervasive challenge in wild deployments characterized by unstable sensor inputs, transmission errors, or restricted access rights.

### 2.2. Recovery-based Methods

To mitigate degradation from data incompleteness, recovery-based paradigms strive to reconstruct missing modalities by conditioning on available observations. Early conventional imputation techniques (Cai et al., 2018; Pan et al., 2022) often struggle with extended missing sequences. Leveraging deep generative capabilities, architectures such as autoencoders (Hinton & Zemel, 1993; Samiee & Kovács, 2023), Generative Adversarial Networks (GANs) (Bischke et al., 2018; Gunasekar et al., 2020; Arya & Saha, 2022), diffusion models (Kingma et al., 2021; Ho et al., 2022; Croitoru et al., 2023; Chen et al., 2023b), and Graph Neural Networks (GNNs) (Gasteiger et al., 2019; Jiang et al., 2019; Xia et al., 2021; Wang et al., 2021; Yu et al., 2023) have been deployed to synthesize missing data. Representative methods include GCNet (Lian et al., 2023), SMIL (Ma et al., 2021), IMDer (Wang et al., 2023b), DiCMoR (Wang et al., 2023a), and MPLMM (Guo et al., 2024). However, the state-of-the-art recovery-based method mainly focuses on reconstructing missing signals, often without explicitly incorporating class-specific statistical priors; moreover, it introduces nontrivial computational overhead in practical deployment.

### 2.3. Non-recovery-based Methods

Alternatively, non-recovery approaches learn robust joint representations directly from incomplete views, employing joint learning (Mahajan et al., 2011; Pham & Venkatesh, 2008) or knowledge distillation (Gou et al., 2021; Cho & Hariharan, 2019; Park et al., 2019; Phuong & Lampert, 2019; Stanton et al., 2021) to handle missing patterns. Representative methods include MCTN (Pham et al., 2019), HME (Zhuang et al., 2025), LNLN (Zhang et al., 2024a), and CorrKD (Li et al., 2024). However, typical non-recovery-based methods use static projections that may weaken affective nuances and instance-specific adaptation.

### 2.4. Prototypical Learning

Prototypical learning (Li et al., 2021; Gupta et al., 2023; Das et al., 2023; Zhang et al., 2024b) typically summarizes each class with representative prototypes in embedding space, where samples are encouraged to approach their corresponding class prototypes and separate from others to establish robust discriminative boundaries. While effective in discrete classification tasks, standard prototypical frameworks are fundamentally ill-posed for sentiment regression, where labels are continuous ($y \in [-3, 3]$) and ordinal relationships must be preserved. Consequently, current literature lacks a mechanism to adapt these discrete prototypes to continuous regression targets without discarding ordinal information.

## 3. The Proposed Method

### 3.1. Preliminaries and Overall Framework

Given an utterance-level sample, we denote its feature set as $\mathcal{X} = \{x^l, x^v, x^a\}$, where $x^m \in \mathbb{R}^{L \times d_m^{\mathrm{orig}}}$ is the sequential feature of modality $m \in \{\text{language}, \text{vision}, \text{audio}\}$, $L$ denotes the input sequence length, and $d_m^{\mathrm{orig}}$ is the original feature dimension. In real-world scenarios, samples often suffer from missing modalities. In this work, we focus on inter-modality missingness, where one or more modalities are entirely missing for a sample, rather than intra-modality missingness, where missingness occurs within frame-level features of an available modality. We denote the set of available modalities for a given sample as $\mathcal{A} \subseteq \{l, v, a\}$, where $\mathcal{A} \neq \emptyset$. The goal is to predict sentiment $y$ from an incomplete multimodal input $\mathcal{X}$ with at least one observed modality available for prediction.

Our framework API couples class-level priors with instance-specific adaptation. We first learn discriminative, class-specific temporal prototypes per modality via prototype-contrastive learning, then impute a missing modality by retrieving its class prototype and adapting it with Directional Instance-Adaptive Affine Modulation (DIAM) conditioned on available modalities. The overview of the proposed model is illustrated in Figure 2.

The design of API addresses two core challenges that motivate SCOPE and DIAM:

**C1 (Semantic Consistency & Efficiency):** How can we better preserve the semantic consistency of imputed features with the target affective class while meeting the low-latency constraints of real-time applications?

**C2 (Instance Adaptation & Efficiency):** How can we dynamically capture instance-specific affective nuances without relying on costly generative processes for imputation?

For each challenge, we propose targeted solutions: Section 3.2 (SCOPE) addresses C1 by constructing explicit

statistical class prototypes, while Section 3.3 (DIAM) tackles C2 via efficient affine modulation.

## 3.2. Semantic-anchored Class-Temporal Prototype Estimation (SCOPE)

**Motivation.** To address **C1**, *SCOPE* constructs non-trainable per-class temporal prototypes as explicit statistical priors. These serve as stable anchors to confine imputed features within a valid semantic space, preventing semantic drift without incurring generative computational overhead.

**Ordinal Sentiment Discretization (OSD).** Standard prototypical learning is typically designed for discrete classification, whereas MSA is a regression task with continuous labels $y \in [-3, 3]$. To bridge this gap, we introduce Ordinal Sentiment Discretization (OSD), which maps continuous scores into $C$ ordered sentiment classes. This adaptation enables the use of class-anchored statistical priors within a regression framework while preserving ordinal sentiment intensities. The choice of $C = 7$ follows the native annotation scheme of CMU–MOSI and CMU–MOSEI, whose sentiment labels use a seven-level Likert-style scale over $[-3, 3]$. Specifically, we discretize continuous sentiment $y \in [-3, 3]$ into $C=7$ classes $c \in \{0, \ldots, 6\}$. The discretization rule is given by:

$$c = \text{clip}\big(\text{round}(y + 3),\, 0,\, 6\big). \tag{1}$$

To obtain the initial feature representations, we first employ a pre-trained BERT encoder for text, followed by 1D temporal convolutional layers to align dimensions. Similarly, raw audio and vision features are projected via distinct 1D temporal convolutions. We denote the resulting projected feature sequence for sample $i$ and modality $m$ as $X_i^m \in \mathbb{R}^{T'_m \times d_m}$, where $d_m$ is the projected hidden dimension and $T'_m$ is the output temporal length after convolution.

For each modality $m$ and class $c$, we maintain a Class-Temporal Prototype Pool (CTPP) with temporal prototypes $P_c^m \in \mathbb{R}^{T'_m \times d_m}$ (stored as non-trainable entries and updated epoch-wise), and per-epoch caches $\{\mathcal{C}_c^m\}$ that collect feature sequences of samples in class $c$. Through OSD and per-modality temporal prototyping, we make class-level priors operational for regression problems while preserving sequential structure and ordinal relationships.

We then compress the temporal dimension to derive a compact semantic descriptor:

$$\mu(X_i^m) = \frac{1}{T'_m} \sum_{t=1}^{T'_m} X_{i,t}^m \in \mathbb{R}^{d_m}, \tag{2}$$

and append these projected features to the corresponding class-specific cache $\mathcal{C}_c^m$. This accumulation constructs an empirical distribution of the class features for the current epoch. To quantify the semantic alignment between the current sample and the global class anchor, we measure the cosine similarity between their respective temporal mean representations:

$$s_c^m(X_i^m) = \cos\big(\mu(X_i^m),\, \mu(P_c^m)\big), \tag{3}$$

where $\mu(P_c^m) = \frac{1}{T'_m} \sum_{t=1}^{T'_m} P_{c,t}^m$.

To encourage semantically discriminative representations, we employ a prototype-contrastive objective. This alignment is applied after the prototypes have been initialized (that is, after the first epoch), when they can provide more meaningful class anchors. For a sample with class $c_i$, we minimize the InfoNCE loss to maximize its similarity to the corresponding class prototype $P_{c_i}^m$ while pushing it away from prototypes of other classes:

$$\ell_i^m = -\log \frac{\exp\big(s_{c_i}^m(X_i^m)/\tau\big)}{\sum_{j \in \mathcal{R}_m} \exp\big(s_j^m(X_i^m)/\tau\big)}, \tag{4}$$

where $\tau$ is the contrastive temperature that controls the concentration of the similarity distribution. The loss is aggregated across available modalities

$$L_{\text{proto}} = \frac{1}{|\mathcal{M}|} \sum_{m \in \mathcal{M}} \frac{1}{N_m} \sum_{i=1}^{N_m} \ell_i^m. \tag{5}$$

At the end of each epoch, we update prototypes independently per modality using the current epoch's data. For class $c$ and modality $m$, we retrieve cached features $\mathcal{S} = \mathcal{C}_c^m$ and apply *Centroid-Filtered Aggregation (CFA)* to refine the distribution. CFA discards the 5% of samples furthest from the class centroid to reduce noise:

$$\mathcal{S}' = \text{TopK}_{\lfloor 0.95|\mathcal{S}| \rfloor} \Big\{ s \in \mathcal{S} : \big\| \mu(s) - \tfrac{1}{|\mathcal{S}|} \sum_{s' \in \mathcal{S}} \mu(s') \big\|_2 \Big\}. \tag{6}$$

We then update the prototype by averaging filtered samples

$$P_c^m = \frac{1}{|\mathcal{S}'|} \sum_{s \in \mathcal{S}'} s \in \mathbb{R}^{T'_m \times d_m}. \tag{7}$$

After aggregation, caches $\mathcal{C}_c^m$ are cleared.

## 3.3. Directional Instance-Adaptive Affine Modulation (DIAM)

**Motivation.** To tackle **C2**, we introduce *Directional Instance-Adaptive Affine Modulation (DIAM)*. Since static prototypes may miss instance-specific nuances (such as intensity variations or speaker-specific prosody), DIAM employs lightweight MLPs for direction-specific affine modulation. This enables precise, sample-level adaptation without the heavy computation of iterative sampling.

*Network structure.* DIAM comprises six direction-specific network groups corresponding to all cross-modal directions (language↔vision, language↔audio, vision↔audio).

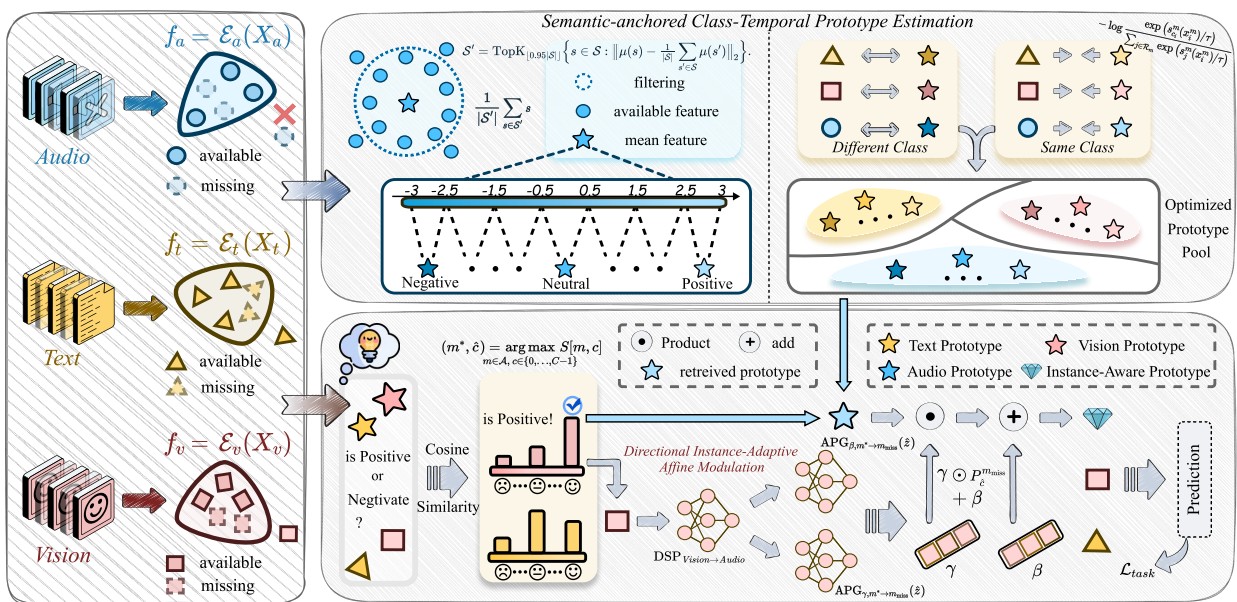

*Figure 2.* **Overview of API framework.** Given multimodal inputs (left), the framework proceeds in two stages. **Top:** Semantic-anchored Class-Temporal Prototype Estimation (SCOPE, Sec 3.2) constructs temporal class prototypes via Centroid-Filtered Aggregation (CFA) to filter outliers, optimizing separability via InfoNCE loss. **Bottom:** Directional Instance-Adaptive Affine Modulation (DIAM, Sec 3.3) identifies the optimal source-class pair via Joint Modality–Class Selection (JMCS) to retrieve the target prototype. It then employs Direction-Specific Projection (DSP) and Affine Parameter Generators (APG-$\gamma$, APG-$\beta$) to produce parameters ($\gamma$, $\beta$) for instance-adaptive modulation, merging class priors with instance cues.

Each group contains three single-hidden-layer MLPs: one Direction-Specific Projection (DSP) that maps the pooled source feature to the missing-modality space, and two Affine Parameter Generators (APG-$\gamma$, APG-$\beta$) that generate feature-wise scaling and shifting parameters.

**Forward process.** Without loss of generality, we detail the imputation for a single missing target modality $m_{\text{miss}}$ given the available set $\mathcal{A}$. DIAM operates in a pipelined manner to produce instance-adaptive imputation:

First, we perform **Joint Modality–Class Selection (JMCS)** to identify the optimal conditioning source. We construct a prototype-similarity grid $S[m, c]$ by computing the cosine similarity between the temporal mean of each available modality $X_i^m$ ($m \in \mathcal{A}$) and all class prototypes $P_c^m$. To prioritize confident modalities, we calculate the class distribution entropy $H^m$ for each modality. The source modality $m^*$ and target class $\hat{c}$ are jointly selected by maximizing similarity:

$$(m^*, \hat{c}) = \underset{m \in \mathcal{A},\, c \in \{0,...,C-1\}}{\arg\max} S[m, c], \qquad (8)$$

breaking ties with minimum entropy $H^m$. Intuitively, this selects the modality–class pair with the highest confidence. During training, $\hat{c}$ is fixed to the ground truth class derived from $y$. The feature–prototype alignment used for inference-time selection is learned by $L_{\text{proto}}$, which encourages features to approach prototypes of their ground-

truth sentiment classes and to separate from prototypes of other classes. Next, we generate adaptive parameters via Direction-Specific Projection (DSP). We pool the selected source feature $X_i^{m^*}$ into a vector $c_{\text{src}}$ and project it into the missing modality's space using a dedicated MLP, yielding $\hat{z} = \text{DSP}_{m^* \to m_{\text{miss}}}(c_{\text{src}})$. This projected representation is then fed into two parallel Affine Parameter Generators (APGs) to produce scaling ($\gamma$) and shifting ($\beta$) vectors:

$$\gamma = \text{APG}_{\gamma, m^* \to m_{\text{miss}}}(\hat{z}), \quad \beta = \text{APG}_{\beta, m^* \to m_{\text{miss}}}(\hat{z}). \quad (9)$$

Finally, we perform Prototype Retrieval and Modulation. We retrieve the static class prototype $P_{\hat{c}}^{m_{\text{miss}}}$ corresponding to the selected class. To inject instance-specific nuances, we modulate this prototype using the generated affine parameters, expanded along the temporal dimension:

$$\tilde{X}_i^{m_{\text{miss}}} = \Gamma \odot P_{\hat{c}}^{m_{\text{miss}}} + B, \qquad (10)$$

where $\Gamma$ and $B$ are the temporally broadcasted versions of $\gamma$ and $\beta$. The resulting instance-adaptive prototype $\tilde{X}_i^{m_{\text{miss}}}$ serves as the imputed feature sequence for the downstream sentiment prediction module.

Overall, DIAM decouples semantic consistency (via prototypes) from instance flexibility (via affine modulation), supporting robust yet expressive imputation. During training, we supervise this process with a reconstruction consistency loss between the instance-adaptive prototypes and the ground-truth features. To encourage the direction-specific

modules to model cross-modal dependencies across directions, we aggregate the loss over all possible source-target pairs $\mathcal{D}_{\text{cross}} = \{(s,t) \mid s,t \in \{l,v,a\}, s \neq t\}$:

$$L_{\text{rec},i} = \frac{1}{|\mathcal{D}_{\text{cross}}|} \sum_{(s,t) \in \mathcal{D}_{\text{cross}}} \frac{1}{T'_t \cdot d_t} \left\| \tilde{X}_i^{t \leftarrow s} - (X_i^t)_{\text{real}} \right\|_F^2, \tag{11}$$

where $\tilde{X}_i^{t \leftarrow s}$ denotes the adaptive prototype representation for target modality $t$ conditioned on source $s$.

### 3.4. Overall Training Objective

We integrate the above losses to reach the full objective:

$$L_{\text{total}} = L_{\text{task}} + \lambda_{\text{proto}} L_{\text{proto}} + \lambda_{\text{rec}} \cdot \frac{1}{B} \sum_{i=1}^{B} L_{\text{rec},i}. \tag{12}$$

Here $L_{\text{task}}$ represents the Mean Absolute Error (MAE) loss for the primary sentiment regression task, $\lambda_{\text{proto}}$ is the contrastive weight that controls the magnitude of class-level separability within each modality, and $\lambda_{\text{rec}}$ is the reconstruction weight that regulates the importance of instance-level fidelity for cross-modal imputation.

## 4. Experiments

We evaluate API by addressing five research questions:

**Q1 (Effectiveness):** Does our method outperform state-of-the-art baselines across varying missing rates? (Sec. 4.1)

**Q2 (Components):** How does each component distinctly contribute to the model's performance? (Sec. 4.2)

**Q3 (Qualitative Mechanism):** Do the visualization results support the intended roles of SCOPE and DIAM in semantic anchoring and instance-adaptive modulation? (Sec. 4.3)

**Q4 (Model Efficiency):** Is the proposed framework more lightweight and efficient compared to representative missing-modality baselines? (Sec. 4.4)

**Q5 (Sensitivity):** How sensitive is API to the coefficients in the training objective? (Sec. 4.5)

**Implementation Details.** Experiments are conducted on two standard benchmarks: CMU–MOSI (Zadeh et al., 2016) (2,199 samples) and CMU–MOSEI (Bagher Zadeh et al., 2018) (22,856 samples). The evaluation uses the official default train/validation/test splits provided by the CMU Multimodal SDK, with 1,284/229/686 samples for CMU–MOSI and 16,326/1,871/4,659 for CMU–MOSEI. Following the experimental setup of HME (Zhuang et al., 2025), we report $ACC_2$ and F1 as evaluation metrics. The text, visual, and audio features are extracted using BERT, Facet, and COVAREP, respectively. All models are implemented in PyTorch and trained on eight NVIDIA RTX 4090 GPUs.

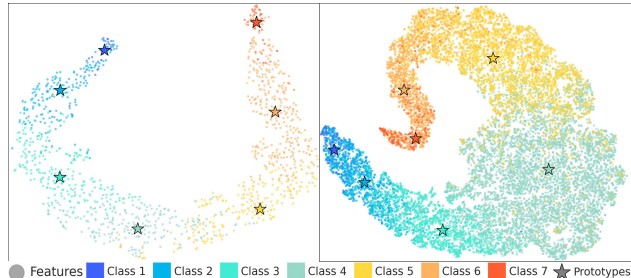

*Figure 3.* t-SNE visualization of text-modality feature and prototype distributions. The upper and lower panels correspond to the MOSI and MOSEI datasets, respectively. Please refer to Sec 4.3.

**Missing Modalities and Random Missing Protocol.** We consider three single-modality missing scenarios—language $\{l\}$, vision $\{v\}$, and audio $\{a\}$—alongside a *random missing* protocol where one or two modalities are randomly masked per sample. To quantify global incompleteness, we follow prior work (Wang et al., 2023b) and employ the *missing rate* (**MR**):

$$\text{MR} = 1 - \frac{\sum_{i=1}^{N} m_i}{N \times M}, \tag{13}$$

where $m_i$ denotes the number of observed modalities for the $i$-th sample, $N$ is the total number of instances, and $M = 3$ is the total modality count. We require $m_i \geq 1$, since a sample with all modalities missing provides no usable input for sentiment prediction, which bounds the missing rate to MR $\leq \frac{M-1}{M} \approx 0.67$. Following IMDer (Wang et al., 2023b), we evaluate our method across MRs $\{0.1, 0.2, \ldots, 0.7\}$, where 0.7 approximates the theoretical ceiling $\frac{M-1}{M}$. The chosen MR remains fixed across all splits.

### 4.1. Comparison with the State-of-the-Art

To address **Q1**, we compare API with state-of-the-art methods in Table 1. The results provide strong evidence for the effectiveness of our approach.

**Overall Performance.** As shown in Table 1, API achieves superior average performance on both datasets. On CMU–MOSI, it obtains an average F1 of 79.6%, outperforming strong baselines such as HME (Zhuang et al., 2025), IMDer+M2AF (Tu et al., 2025), and LNLN (Zhang et al., 2024a). This demonstrates the robustness of our framework regardless of the severity of data incompleteness.

**Resilience to Extreme Missingness.** The resilience of API is particularly evident on the MOSI dataset under extreme conditions (MR=0.7), where it achieves a 74.1% F1 score, significantly surpassing GCNet (65.4%) and MPLMM (56.2%). This resilience suggests that API can maintain relatively stable inference even when valid signals are severely limited. On the larger MOSEI dataset, our method maintains competitive performance comparable to

*Table 1.* Results on CMU–MOSI and CMU–MOSEI under varying missing rates (MR). We report ACC$_2$ and F1 (%). Higher values indicate better performance for both metrics. Bold indicates the best per MR. ★ denotes results reported in HME (Zhuang et al., 2025). † denotes results reported in M2AF (Tu et al., 2025). See details in Sec 4.1.

| Datasets | MR | GCNet★ | | MPLMM★ | | DiCMoR★ | | IMDer★ | | IMDer+M2AF† | | LNLN★ | | HME★ | | API (Ours) | |
|---|---|---|---|---|---|---|---|---|---|---|---|---|---|---|---|---|---|
| | | ACC$_2$ | F1 | ACC$_2$ | F1 | ACC$_2$ | F1 | ACC$_2$ | F1 | ACC$_2$ | F1 | ACC$_2$ | F1 | ACC$_2$ | F1 | ACC$_2$ | F1 |
| | 0.1 | 82.3 | 82.3 | 80.6 | 80.7 | 83.2 | 83.2 | 84.6 | 84.4 | 81.6 | 81.7 | 81.4 | 81.4 | 84.9 | 84.7 | **85.4** | **85.3** |
| | 0.2 | 79.4 | 79.5 | 77.7 | 77.9 | 81.6 | 81.3 | 81.7 | 81.8 | 76.8 | 76.9 | 78.7 | 78.7 | 82.9 | 82.9 | **83.2** | **83.2** |
| | 0.3 | 77.2 | 77.2 | 75.0 | 75.1 | 78.4 | 77.9 | 79.9 | 79.4 | 78.5 | 78.6 | 74.7 | 74.8 | 81.1 | 81.0 | **81.5** | **81.7** |
| MOSI | 0.4 | 74.3 | 74.4 | 70.0 | 69.8 | 76.2 | 74.7 | 78.1 | 77.2 | 68.3 | 67.6 | 70.3 | 69.9 | 79.9 | 80.0 | **80.3** | **80.4** |
| | 0.5 | 70.0 | 69.8 | 67.4 | 66.6 | 72.6 | 72.7 | 73.6 | 73.7 | 64.0 | 62.1 | 67.1 | 66.3 | 76.4 | 76.4 | **76.8** | **76.8** |
| | 0.6 | 67.7 | 66.7 | 63.4 | 61.8 | 71.3 | 71.4 | 72.9 | 69.8 | 65.8 | 65.3 | 62.5 | 60.5 | 74.5 | 74.4 | **75.8** | **75.7** |
| | 0.7 | 65.7 | 65.4 | 59.2 | 56.2 | 69.1 | 69.2 | 67.7 | 67.1 | 62.2 | 61.7 | 61.0 | 58.3 | 73.5 | 71.7 | **73.9** | **74.1** |
| | Avg. | 73.8 | 73.6 | 70.5 | 69.7 | 76.1 | 75.8 | 76.9 | 76.2 | 71.0 | 70.6 | 70.8 | 70.0 | 79.0 | 78.7 | **79.6** | **79.6** |
| | 0.1 | 82.3 | 82.1 | 83.7 | 83.5 | 83.2 | 83.1 | 83.5 | 83.3 | **84.3** | **84.3** | 83.1 | 83.0 | 84.0 | 83.8 | 84.1 | 84.1 |
| | 0.2 | 80.3 | 79.9 | 81.4 | 80.8 | 81.2 | 80.8 | 81.3 | 80.9 | 81.2 | 80.7 | 81.7 | 81.0 | 82.3 | 81.9 | **82.9** | **82.8** |
| | 0.3 | 77.5 | 76.8 | 79.2 | 78.3 | 79.0 | 77.9 | 79.4 | 78.8 | 79.1 | 78.1 | 79.1 | 78.1 | 80.0 | 79.6 | **80.9** | **81.1** |
| MOSEI | 0.4 | 76.0 | 74.9 | 77.1 | 76.1 | 76.9 | 75.2 | 76.1 | 74.3 | 77.4 | 76.0 | 76.9 | 75.6 | 77.6 | 76.4 | **78.5** | **78.3** |
| | 0.5 | 74.9 | 73.2 | 75.6 | 73.8 | 73.7 | 71.7 | 75.3 | 72.4 | 74.6 | 73.4 | 75.1 | 72.4 | **75.7** | 74.7 | 75.6 | **75.3** |
| | 0.6 | 74.1 | 72.1 | 72.7 | 70.6 | 71.1 | 70.4 | 71.2 | 66.5 | 71.9 | 68.7 | 72.5 | 68.6 | 73.0 | 72.5 | **74.2** | **73.7** |
| | 0.7 | **73.2** | **70.4** | 70.5 | 69.2 | 70.6 | 68.9 | 71.1 | 65.9 | 70.6 | 66.7 | 70.6 | 65.2 | 72.4 | 69.2 | 72.4 | 69.3 |
| | Avg. | 76.9 | 75.6 | 77.2 | 76.0 | 76.5 | 75.4 | 76.8 | 74.6 | 77.0 | 75.4 | 77.0 | 74.8 | 77.9 | 76.9 | **78.4** | **77.8** |

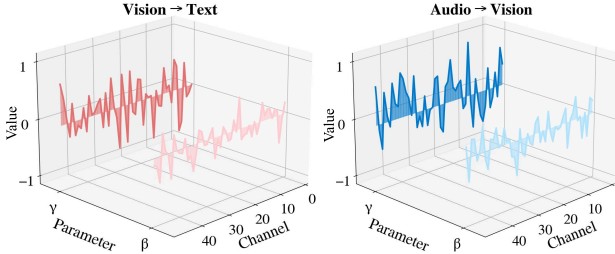

*Figure 4.* Visualization of Joint Modality–Class Selection (JMCS) under three missing cases. From top to bottom: vision missing, text missing, and audio missing. Please refer to Sec 4.3.

*Figure 5.* Visualization of scaling ($\gamma$) and shifting ($\beta$) parameters generated by DIAM. The panels contrast modulation directions for Vision → Text (left) and Audio → Vision (right). Refer to Sec 4.3.

top baselines at high missing rates.

**Performance Across Datasets of Different Scales.** API is evaluated on both CMU–MOSI and CMU–MOSEI, covering benchmark datasets of different scales. On CMU–MOSI, it achieves the best average ACC$_2$ and F1 across missing rates. On CMU–MOSEI, it also obtains the best average performance among the compared methods. These results indicate that the proposed framework remains competitive across both benchmarks.

### 4.2. Component Analysis

To address **Q2**, we dissect the contribution of each component within API via an ablation study using an incremental validity strategy, as summarized in Table 2. The baseline employs a Zero-Padding strategy where missing modalities are simply replaced with zero vectors.

**Incremental Evaluation Design.** Since DIAM functions by modulating the prototypes retrieved via SCOPE, it is struc-turally dependent on the latter. Consequently, we evaluate our framework in a strictly incremental order: starting from the *Baseline*, adding *SCOPE*, and finally integrating *DIAM* to form the complete API.

**Effect of SCOPE (Class Priors).** Integrating SCOPE yields a substantial performance gain (+4.7% F1 on MOSI) over the baseline. By introducing non-trainable class prototypes derived from the momentum encoder, SCOPE provides stable semantic anchors. This improvement corroborates that leveraging class-level statistical priors offers a significantly more robust semantic initialization than zero-padding, even in the absence of learnable generation parameters.

**Effect of DIAM (Instance Modulation).** The complete API framework (adding DIAM) further elevates performance, increasing F1 from 76.9% to 79.6% on MOSI. While SCOPE mitigates semantic drift via static anchors, it lacks instance-level granularity. DIAM bridges this gap by applying dynamic affine transformations to these anchors. This result suggests that tailoring general class priors with instance-specific modulation improves prediction reliability across

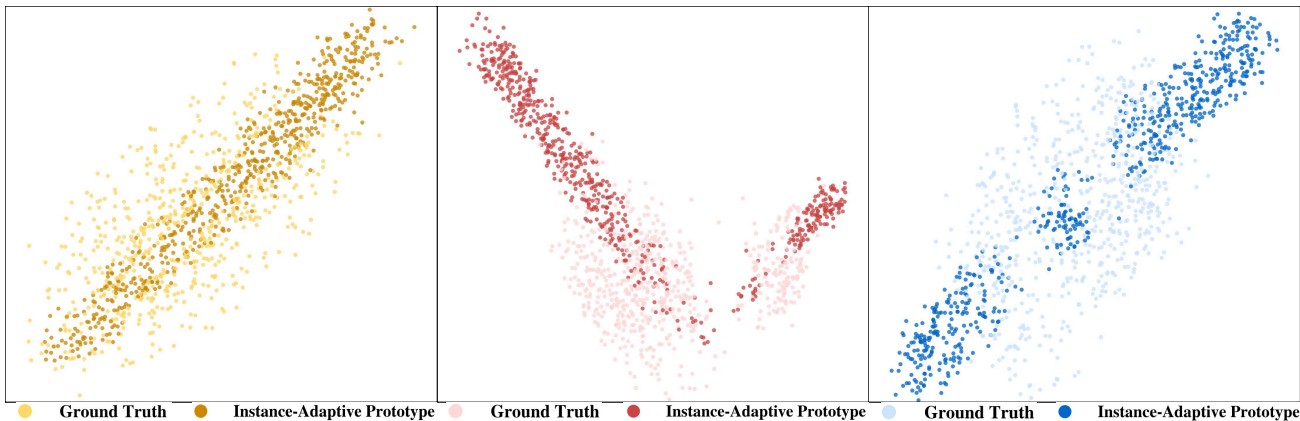

*Figure 6.* t-SNE visualization comparing the distribution alignment between imputed instance-adaptive prototypes and ground-truth features. From left to right, the panels represent scenarios where Text, Audio, and Vision are the missing modalities. Refer to Sec 4.3.

*Table 2.* Ablation study of the key components in API on MOSI and MOSEI datasets. Bold indicates the best performance. ↑ denotes the improvement over the baseline. Please refer to Sec 4.2.

| Dataset | SCOPE | DIAM | $ACC_2$ | F1 |
|---------|-------|------|---------|-----|
| MOSI | | | 72.2 | 72.2 |
| | ✓ | | $76.9_{\uparrow 4.7}$ | $76.9_{\uparrow 4.7}$ |
| | ✓ | ✓ | $\mathbf{79.6}_{\uparrow 7.4}$ | $\mathbf{79.6}_{\uparrow 7.4}$ |
| MOSEI | | | 73.4 | 72.4 |
| | ✓ | | $75.4_{\uparrow 2.0}$ | $74.0_{\uparrow 1.6}$ |
| | ✓ | ✓ | $\mathbf{78.4}_{\uparrow 5.0}$ | $\mathbf{77.8}_{\uparrow 5.4}$ |

missing-rate conditions.

### 4.3. Mechanism Analysis

To answer **Q3**, we visualize feature distributions and modulation parameters, checking whether SCOPE behaves as a semantic anchoring mechanism and DIAM provides the expected instance-conditioned calibration.

**Visualization of Semantic Anchoring.** To analyze whether SCOPE provides useful class-level anchors, we visualize the distributions of text-modality features and learned prototypes using t-SNE in Figure 3. This visualization is derived from one training epoch under a 30% missing rate. As observed, the learned prototypes (stars) show clear *inter-class separability* and are surrounded by features (points) from the corresponding sentiment classes. This pattern is consistent with SCOPE capturing discriminative class-level priors as semantic anchors for imputation.

**Visualization of Joint Modality–Class Selection.** We visualize the prototype selection confidence in Figure 4. The heatmap illustrates the matching scores between available modalities and class prototypes, where highlighted boxes indicate the selected source-class pairs used for imputation.

**Visualization of Instance-Specific Adaptation.** We visualize the affine parameters ($\gamma, \beta$) in Figure 5 to assess whether

DIAM produces sample-dependent modulation patterns rather than a fixed bias. The observed variations across samples and modulation directions are consistent with DIAM introducing direction-specific and instance-adaptive adjustments to the retrieved prototypes.

**Distributional Alignment of Imputation.** To assess the distributional behavior of the imputed representations, we randomly select samples from the CMU-MOSEI testing set and visualize the overlap between the *Instance-Adaptive Prototypes* of the missing modality and their corresponding *Ground-Truth* features in Figure 6. The visualization shows that the imputed prototypes tend to occupy similar semantic regions as the ground-truth features, providing qualitative support for the compatibility between prototype retrieval and instance-adaptive modulation.

### 4.4. Efficiency Analysis

To address **Q4**, we benchmark the efficiency of API against representative missing-modality baselines, including IMDer (Wang et al., 2023b), HME (Zhuang et al., 2025), and LNLN (Zhang et al., 2024a). For a comprehensive evaluation, we report learnable parameters, FLOPs, and inference latency. Specifically, *Miss.* denotes learnable parameters associated with missing-modality handling, while FLOPs and latency are averaged over all scenarios within each category: 1-Miss (mean of $\{l\}, \{v\}, \{a\}$ missing) and 2-Miss (mean of $\{l, v\}, \{l, a\}, \{v, a\}$ missing). Table 3 confirms our framework's efficiency advantage.

**Lightweight Architecture.** Our framework is designed for minimal architectural redundancy. As SCOPE relies on non-trainable statistical prototypes, it introduces zero learnable parameters. Consequently, the missing-modality parameter overhead is exclusively attributed to DIAM, which requires negligible capacity (∼0.14M on MOSI and ∼0.05M on MO-SEI). This is lower than IMDer, HME, and LNLN, whose

*Table 3.* Efficiency comparison on MOSI and MOSEI. *Miss.* denotes learnable parameters associated with missing-modality handling. We also report total parameters, FLOPs, and inference latency.

| Dataset | Method | Params (M) | | FLOPs (G) | | Latency (ms) | |
|---------|--------|------|-------|--------|--------|--------|--------|
| | | Miss. | Total | 1-Miss | 2-Miss | 1-Miss | 2-Miss |
| MOSI | IMDer | 11.95 | 122.48 | 14.37 | 20.16 | 2241 | 4386 |
| | HME | 36.43 | 152.98 | 18.67 | 18.67 | 951 | 974 |
| | LNLN | 2.99 | 115.97 | 11.39 | 11.39 | 65 | 66 |
| | API | **0.14** | **112.25** | **8.67** | **8.67** | **46** | **48** |
| MOSEI | IMDer | 11.95 | 122.94 | 14.40 | 20.19 | 2443 | 4389 |
| | HME | 31.69 | 145.59 | 18.03 | 18.03 | 933 | 834 |
| | LNLN | 2.99 | 116.58 | 12.01 | 12.01 | 56 | 65 |
| | API | **0.05** | **110.54** | **8.57** | **8.57** | **47** | **46** |

missing-modality handling components require millions of learnable parameters.

**Stable Computational Cost.** A pivotal advantage of API is its computational stability. As the number of missing modalities increases, IMDer's FLOPs escalate linearly (surging by +5.8G per view) due to generative sampling. In contrast, API incurs negligible overhead: specifically, filling one missing view adds only 0.045 MFLOPs on MOSI and 0.016 MFLOPs on MOSEI. These microscopic increases ($< 0.001\%$) keep the total computational footprint stable ($\sim$8.6G); as shown in Table 3, API also uses fewer FLOPs than IMDer, HME, and LNLN on both MOSI and MOSEI under 1-Miss and 2-Miss scenarios.

**Real-Time Inference Speed.** Latency metrics further corroborate the deployment viability of API. On the large-scale MOSEI dataset, our inference time stays around $\sim$47ms under missing scenarios, as the computational cost of DIAM is so minimal that it falls within system variance. Compared with HME and LNLN, API also achieves lower latency while using fewer missing-modality-related parameters. IMDer suffers from prohibitive latency penalties, spiking to over 2000ms (one missing view) and 4000ms (two missing views) due to iterative sampling. Consequently, API achieves substantial speedups over IMDer on both MOSI and MOSEI, while retaining strong performance under missing-modality settings across the two datasets.

### 4.5. Sensitivity Analysis

To address **Q5**, we investigate the sensitivity of API to two key coefficients in the training objective, $\lambda_{\text{proto}}$ and $\lambda_{\text{rec}}$, on the MOSI dataset under a 30% missing rate (Figure 7). The former controls the strength of the feature–prototype contrastive constraint, while the latter controls the reconstruction constraint applied to imputed representations. In the sensitivity analysis, we vary the value of one hyperparameter while keeping the other hyperparameters fixed to the settings used in the main experiments. As shown in

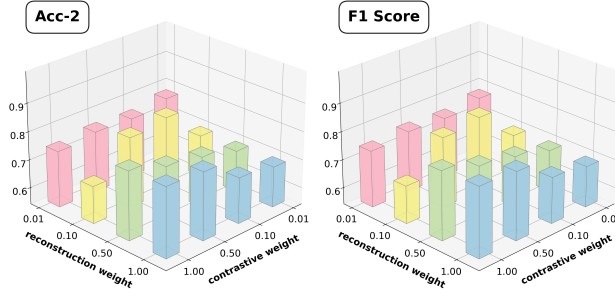

*Figure 7.* Sensitivity analysis on the MOSI dataset at a 30% missing rate. Refer to Sec. 4.5.

Figure 7, both $\text{ACC}_2$ and F1 reach their best values around $\lambda_{\text{proto}} = 0.1$ and $\lambda_{\text{rec}} = 0.1$, supporting our default configuration. When either coefficient is too small, the corresponding feature–prototype contrastive constraint or reconstruction constraint becomes insufficient. Conversely, overly large coefficients may overemphasize auxiliary objectives and weaken the primary sentiment prediction objective, leading to performance degradation. Overall, the performance remains relatively stable across a broad range around the default setting, suggesting that API does not depend on highly precise hyperparameter tuning.

## 5. Conclusion

This paper presents API, a prototype-centric framework for incomplete MSA. By synergizing SCOPE's class-level anchors with DIAM's instance-adaptive modulation, our approach promotes semantic consistency and captures sample-specific nuances while maintaining low inference latency. Experiments on CMU–MOSI and CMU–MOSEI show that API achieves competitive predictive performance under varying missing rates while introducing only lightweight missing-modality handling overhead, supporting its practicality for incomplete MSA settings.

**Limitations.** This work focuses on utterance-level inter-modality missingness; extending API to intra-modality missingness is left to future work.

**Acknowledgements.** This work is supported by the National Key Research and Development Program of China (2024YFC3308400), the National Natural Science Foundation of China under Grants T2541022 and 62501428, the Postdoctoral Fellowship Program of China Postdoctoral Science Foundation (GZC20241268), and the Hubei Provincial Natural Science Foundation of China (2025AFB219).

## Impact Statement

This work studies incomplete multimodal sentiment analysis under missing modalities. Risks include demographic bias, privacy concerns, and misinterpretation of affective states. Such systems should be used cautiously in high-stakes settings and not treated as definitive psychological assessments or sole decision criteria.

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
