# OpenReview forum: "API: Adaptive Prototype Imputation for Incomplete Multimodal Sentiment Analysis"
_ICML.cc/2026/Conference — ICML 2026 regular_

### Official Review · Reviewer_jMTd · 2026-03-11

**Soundness:** 3
**Presentation:** 4
**Significance:** 3
**Originality:** 3
**Overall Recommendation:** 4
**Confidence:** 4

**Summary:**

This paper addresses the problem of missing modalities in MSA. This paper proposes an Adaptive Prototype Imputation (API) framework to overcome two limitations of existing approaches: (1) computational inefficiency and semantic inconsistency, and (2) insufficient instance specificity. To address these challenges, the paper introduces two components: Semantic-anchored Class-Temporal Prototype Estimation (SCOPE) and Directional Instance-Adaptive Affine Modulation (DIAM). Experiments on the CMU-MOSI and CMU-MOSEI datasets demonstrate the effectiveness of the proposed API framework. The results also support the contributions of the individual modules, the overall design, and the efficiency of the approach.

**Compliance With Llm Reviewing Policy:**

Affirmed.

**Final Justification:**

The authors’ rebuttal addresses most of my concerns, and I appreciate the additional clarifications provided. However, in the newly added experiments on the IEMOCAP dataset, the experimental settings appear to differ from those used by the selected baseline, HME. In particular, differences in feature extraction methods can have a significant impact on multimodal sentiment analysis performance. For a fair and meaningful comparison, it would be preferable to evaluate all methods under consistent feature extraction settings.

Given this concern, I maintain my initial score. Overall, I consider the paper to be a weak accept.

**Key Questions For Authors:**

1. Missing modalities in real-world applications can occur in various forms. However, the current evaluation focuses primarily on random modality missingness. Could the authors comment on how API performs under other commonly studied scenarios, such as fixed-modality missingness (as discussed in HME) or intra-modality missingness (as explored in CorrKD)?

2. The proposed SCOPE module seems related to the Category-guided Prototype Distillation method used in CorrKD, where class-specific prototypes are constructed by averaging representations of samples from the same category and then used for alignment. Could the authors clarify the main differences between these two approaches?

3. The paper aims to ensure semantic consistency while maintaining low-latency inference, but Table 3 only compares efficiency with IMDER, which uses a diffusion-based reconstruction process. Could the authors provide additional efficiency comparisons with other baselines such as HME, and LNLN, which use lightweight reconstruction mechanisms?

4. In Line 268, the paper states that all models are trained on **eight NVIDIA RTX 4090 GPUs**, which appears to be more computationally demanding than some related work. For example, HME reports that training and inference can be completed on a single RTX 3090 GPU. Could the authors provide additional comparisons of training time and inference time with models such as LNLN and HME to better support the claim that API is lightweight? In addition, since the proposed framework uses contrastive learning, it would be helpful to understand the trade-off between batch size and GPU memory usage, as this may affect the practical applicability of the method.

**Limitations:**

yes

**Strengths And Weaknesses:**

This work provides a clear analysis of the missing-modality problem in MSA and formulates two key research questions: (1) how to ensure that imputed features remain semantically consistent with the target affective class while maintaining the low-latency requirements of real-time applications, and (2) how to capture instance-specific affective nuances without relying on computationally expensive generative models. The proposed modules are designed specifically to address these two questions. Overall, the paper is well organized, clearly written, and the proposed method appears effective.

However, I have several concerns and suggestions that may help further improve the work.

First, missing modalities in real-world scenarios can occur in many forms. However, the paper appears to focus mainly on random modality missingness. Other commonly studied scenarios are not considered. For example, the fixed-modality missing setting discussed in HME and the intra-modality missingness scenario explored in CorrKD are not evaluated. Including additional missing-modality settings would provide a more comprehensive evaluation and help demonstrate the robustness of the proposed API framework.

Second, the paper focuses only on regression-based sentiment prediction. In practice, many affective tasks involve emotion classification, which is also an important setting for multimodal affect analysis. Prior work such as HME and CorrKD has validated their methods on the IEMOCAP dataset. Extending the experiments to include emotion recognition tasks would help demonstrate the broader applicability of API.

Third, there are several unclear or undefined notations in the paper. For example, in Line 149, the meaning of $L \times d_m$​ is not clearly explained. Similarly, the variable $T'_m$​ in Line 151 is not defined, and its possible values are unclear. In addition, Figure 1 is not clearly referenced or explained in the main text. Clarifying these details would improve the readability of the paper.

Fourth, the proposed SCOPE module appears conceptually related to the Category-guided Prototype Distillation mechanism used in CorrKD (baseline in the paper). Both methods compute class-specific representations by aggregating features from samples of the same category and then align the current sample representation with these class prototypes. It would be helpful if the authors could further clarify the key conceptual or technical differences between SCOPE and the prototype learning strategy used in CorrKD.

Fifth, the description of the cross-dataset generalization experiment in Line 311 is somewhat ambiguous. The current wording may give readers the impression that the model is trained on one dataset and directly tested on another dataset. If this is not the case, the authors may consider clarifying the description. Alternatively, performing an explicit cross-dataset experiment could strengthen the evaluation.

Sixth, one of the main motivations of the paper is to ensure semantic consistency of imputed features while maintaining low-latency inference. However, in Table 3, the efficiency comparison only includes IMDER, which relies on a diffusion model. It would be more informative to also compare with other representative baselines such as GCNet, HME, and LNLN. These methods typically reconstruct missing modalities using lightweight architectures such as MLPs or prompts and are therefore relatively efficient. Including such comparisons would provide a more convincing demonstration of the efficiency advantages of the proposed API method.

---

> ### Author Rebuttal · Authors · 2026-03-31
>
> Dear Reviewer jMTd
>
> Thank you for the careful and constructive review. It is especially encouraging to hear that the paper is viewed as well organized, clearly written, and that the proposed method appears effective. The helpful suggestions are also greatly appreciated, and we address each point below.
>
> Here, `Q` denotes `Question` and `S` denotes `Suggestion`.
>
> > `Q1 & S1`: Evaluation beyond random missingness
>
> We additionally evaluate API under the fixed-modality missingness setting on MOSI, MOSEI, and IEMOCAP.The results are as follows.
>
> | Dataset | Avail | **API (ACC2/F1)** | HME (ACC2/F1) |
> | --- | --- | ---: | ---: |
> | IEMOCAP | L | **83.9/84.0** | 79.1/78.1 |
> |  | V | **77.2/77.1** | 75.7/70.8 |
> |  | A | **80.0/79.6** | 78.9/76.8 |
> |  | L,V | **86.1/86.1** | 79.7/78.9 |
> |  | L,A | **86.4/86.4** | 80.7/80.0 |
> |  | A,V | **80.2/79.9** | 78.1/76.3 |
> | MOSI | L | **85.8/85.8** | 83.9/83.8 |
> |  | V | **64.5/64.4** | 57.8/54.6 |
> |  | A | **63.3/63.2** | 57.8/57.7 |
> |  | L,V | **86.1/86.0** | 84.3/84.3 |
> |  | L,A | **86.3/86.1** | 84.4/84.4 |
> |  | A,V | **65.2/65.2** | 57.8/55.5 |
> | MOSEI | L | **85.5/85.4** | 84.7/84.7 |
> |  | V | **66.7/65.4** | 62.9/53.0 |
> |  | A | **65.1/65.5** | 63.0/50.4 |
> |  | L,V | **85.6/85.6** | 84.8/84.8 |
> |  | L,A | **85.7/85.5** | 84.4/84.4 |
> |  | A,V | **66.1/65.0** | 62.9/55.6 |
>
>
> > `Q2 & S4`: Difference from CorrKD's category-guided prototype distillation.
>
> While both methods use category-level prototypes, their roles are fundamentally different. CorrKD is a teacher-student distillation framework in which a teacher trained on complete-modality samples provides joint multimodal prototypes, and these prototypes serve only as a training-time signal for aligning teacher-student similarity matrices. In contrast, our method maintains modality-specific temporal prototypes and uses them as direct imputation anchors at both training and inference.
>
>
>
> > `Q3 & S6`: Additional efficiency comparisons.
>
> We additionally evaluated HME and LNLN. Here, `Miss.` denotes the learnable parameters specifically associated with missing-modality handling, `Total` the total learnable parameters, and `Lat.` the inference latency. The results are as follows.
>
> | Dataset | Method | Miss. (M) | Total (M) | FLOPs 1-Miss (G) | FLOPs 2-Miss (G) | Lat. 1-Miss (ms) | Lat. 2-Miss (ms) |
> | --- | --- | ---: | ---: | ---: | ---: | ---: | ---: |
> |  | HME | 36.43 | 152.98 | 18.67 | 18.67 | 951 | 974 |
> | MOSI | LNLN | 2.99 | 115.97 | 11.39 | 11.39 | 65 | 66 |
> |  | **API** | **0.14** | **112.25** | **8.67** | **8.67** | **46** | **48** |
> |  | HME | 31.69 | 145.59 | 18.03 | 18.03 | 933 | 834 |
> | MOSEI | LNLN | 2.99 | 116.58 | 12.01 | 12.01 | 56 | 65 |
> |  | **API** | **0.05** | **110.54** | **8.57** | **8.57** | **47** | **46** |
>
> > `Q4`: Single-GPU runtime and memory usage.
>
> The `8 × RTX 4090` statement describes our available environment, not the hardware requirement of API. API can be trained and inferred on a single RTX 3090 GPU.
>
> On MOSI, we additionally evaluate training time, comparisons of inference latency, and the trade-off between batch size and GPU memory usage. Comparisons of inference latency are reported in Q3 & S6; the results below report training time and batch-size/memory trade-off, where ACC2/F1 is averaged over all missing rates (MR=0.1-0.7).
>
> | Method | Training Time (30 epochs, s) |
> | --- | ---: |
> | **API** | **1080** |
> | LNLN | 2746 |
> | HME | 3378 |
>
> | Batch Size | ACC2/F1 | Peak GPU Memory (GB) |
> | --- | ---: | ---: |
> | 16 | 75.6/75.7 | 2.46 |
> | 32 | 77.0/77.1 | 3.36 |
> | 64 | 79.6/79.6 | 4.59 |
> | 128 | 79.0/79.0 | 7.45 |
>
>
> > `S2`: Additional dataset.
>
> We further evaluate API on IEMOCAP under the random-missing protocol. The results are as follows.
>
> | mr | **API (ACC2/F1)** | HME (ACC2/F1) |
> | --- | ---: | ---: |
> | 0.1 | **85.2/85.2** | 80.9/80.3 |
> | 0.2 | **83.5/83.4** | 80.7/80.0 |
> | 0.3 | **82.9/82.8** | 79.0/77.8 |
> | 0.4 | **81.1/81.0** | 78.7/77.3 |
> | 0.5 | **79.5/79.5** | 78.5/77.1 |
> | 0.6 | **79.0/78.9** | 77.8/76.1 |
> | 0.7 | **77.9/78.0** | 77.6/76.0 |
>
>
>
> > `S3`: Clarification of notation and Figure 1.
>
> There is a typo in the notation: the first $d_m$ should be $d_m^{\mathrm{orig}}$. Specifically, in $x^m \in \mathbb{R}^{L \times d_m^{\mathrm{orig}}}$, $L$ denotes the input sequence length, while $d_m^{\mathrm{orig}}$ denotes the original feature dimension of modality $m$. After the modality-specific 1D temporal convolution, the feature sequence becomes $X_i^m \in \mathbb{R}^{d_m \times T'_m}$, where $d_m$ denotes the projected hidden dimension and $T'_m$ is the output temporal length after convolution. We will revise the notation and define $L$ and $T'_m$ on first use.
>
> We will cite Fig. 1 explicitly in the Introduction.
>
> > `S5`: Clarification of the cross-dataset description.
>
> This is not a cross-dataset transfer setting. Rather, MOSI and MOSEI are trained and evaluated independently using their standard splits. We will rename this to ''Performance Across Datasets of Different Scales''.

---

> > ### Author Rebuttal · Reviewer_jMTd · 2026-04-02
> >
> > Thank you to the authors for the detailed and thoughtful response. I still have a few remaining questions regarding the additional experiments on the IEMOCAP dataset, which show very strong performance.
> >
> > Could the authors please provide more details about the experimental setup? In particular, it would be helpful to clarify the feature dimensions used for each of the three modalities, the data preprocessing pipeline, and the key hyperparameters. Additionally, are these settings consistent with those used in prior work, or were any modifications made specifically for this study?
> >
> > Including this information would improve transparency and reproducibility, and further strengthen the evidence supporting the effectiveness of the proposed method.

---

> > > ### Author Response · Authors · 2026-04-03
> > >
> > > Dear Reviewer jMTd,
> > >
> > > Thank you again for the thoughtful follow-up and for recognizing the strong performance of API on IEMOCAP. We also appreciate this helpful request for additional implementation details and provide a more explicit description of the IEMOCAP setup below.
> > >
> > > > `Follow-up question`: Additional details on the IEMOCAP setup
> > >
> > > For clarity, we summarize the IEMOCAP setup for the three modalities below:
> > >
> > > - Acoustic: **130**-dimensional ComParE features extracted with OpenSMILE under the IS13 ComParE configuration.
> > > - Visual: **342**-dimensional DenseFace features extracted from detected face frames using a DenseNet pretrained on FER+.
> > > - Textual: **1024**-dimensional contextual word embeddings extracted using a pretrained BERT-large model.
> > >
> > > These settings are consistent with MPLMM, which follows a standard and explicitly documented feature extraction pipeline for the IEMOCAP benchmark.
> > >
> > > The key hyperparameters used in the IEMOCAP experiments are summarized as follows:
> > >
> > > | Hyperparameter | Value |
> > > | --- | ---: |
> > > | Learning rate | `1e-4` |
> > > | Batch size | `128` |
> > > | Contrastive weight | `0.1` |
> > > | Reconstruction weight | `0.1` |
> > >
> > > In addition, all reported IEMOCAP results are averaged over five random seeds. We will make these details explicit in the final version to further improve transparency and reproducibility. To further support reproducibility, the code will also be released so that the full implementation details are readily accessible.
> > >
> > > > `Clarification on Feature Extraction Consistency`
> > >
> > > We respect your concern regarding consistent feature extraction and would like to clarify that our experimental setup strictly follows the evaluation precedent established by HME itself.
> > >
> > > **1. Following HME’s Own Evaluation Precedent:**
> > >
> > > In **Table 1** of the HME paper (IEMOCAP fixed missing protocol), HME directly cited MPLMM's originally published results. Since HME treats MPLMM as a valid comparison target despite the differences in feature extraction, it explicitly validates this comparison as fair and meaningful. Therefore, by evaluating our API using the feature extraction setup consistent with MPLMM and comparing it with HME, we are strictly adhering to the exact paradigm established by HME itself.
> > >
> > > **2. Ensuring Reproducibility:**
> > >
> > > We adopted a feature extraction setup consistent with MPLMM because it is fully open-sourced, highly standardized, and widely verified by the community. This ensures our experimental results and the API framework remain highly transparent and reproducible.
> > >
> > > We hope this clarifies that our experimental design was chosen to ensure reproducibility, while remaining consistent with the evaluation standards accepted by HME.

---

### Official Review · Reviewer_LkTn · 2026-03-12

**Soundness:** 3
**Presentation:** 3
**Significance:** 3
**Originality:** 3
**Overall Recommendation:** 4
**Confidence:** 4

**Summary:**

This paper proposes Adaptive Prototype Imputation (API), a prototype-centric framework for multimodal sentiment analysis (MSA) under missing modality conditions. The work addresses the common real-world scenario where one or more modalities (text, audio, or vision) are unavailable due to sensor failures, privacy restrictions, or transmission errors. A fundamental problem presented by this paper is how to impute missing modality representations that are both semantically consistent with the target sentiment class and adaptive to instance-level nuances, while remaining computationally efficient for real-time inference.

To address this problem, the authors introduce the API framework, which consists of two main components:
1. Semantic-anchored Class-Temporal Prototype Estimation (SCOPE)
SCOPE constructs non-trainable class-specific temporal prototypes for each modality by aggregating features across training samples. Continuous sentiment scores are discretized into ordinal classes, enabling the use of class prototypes as semantic anchors. These prototypes constrain the imputed representations to remain within the appropriate semantic space, reducing the risk of semantic drift.
2. Directional Instance-Adaptive Affine Modulation (DIAM)
DIAM adapts retrieved prototypes to individual samples using direction-specific affine transformations conditioned on available modalities. It first selects the most confident modality–class pair, then generates scaling and shifting parameters via lightweight MLPs to modulate the prototype of the missing modality, producing instance-adaptive imputed features.

Experiments on the CMU-MOSI and CMU-MOSEI datasets show that API consistently outperforms several state-of-the-art incomplete-MSA methods across various missing-modality rates while maintaining significantly lower inference latency and parameter overhead. Efficiency analyses further demonstrate that the proposed method achieves orders-of-magnitude speed improvements compared with diffusion-based recovery approaches.

**Compliance With Llm Reviewing Policy:**

Affirmed.

**Final Justification:**

The author's reply was very detailed and resolved most of my doubts.

**Key Questions For Authors:**

1. Generalization beyond sentiment analysis tasks
The experimental evaluation is limited to the CMU-MOSI and CMU-MOSEI datasets, both of which focus on multimodal sentiment analysis. How well does the proposed API framework generalize to other multimodal tasks with missing modalities (e.g., emotion recognition in conversations, multimodal action recognition, or multimodal reasoning tasks)?

2. Effect of ordinal discretization on regression performance
The method discretizes continuous sentiment labels y∈[−3,3] into seven ordinal classes to construct class prototypes (OSD). Have the authors analyzed how sensitive the performance is to this discretization scheme (e.g., different numbers of bins or continuous prototype learning)?

3. Applicability to scenarios with multiple simultaneously missing modalities
The paper reports experiments under varying missing rates and random masking. However, could the authors clarify how the framework behaves in extreme scenarios where two modalities are consistently missing and only one modality remains available?

**Limitations:**

No.

The paper includes an Impact Statement, but it briefly states that the work advances machine learning and that there are no societal consequences requiring discussion. This response is insufficient, particularly for a paper addressing multimodal sentiment analysis and affect recognition.

Suggested improvements

1. Address robustness and deployment limitations.
In real-world systems, missing modalities may occur in structured patterns (e.g., consistently missing sensors) rather than random masking. The authors should discuss whether the proposed prototype-based imputation approach remains reliable under such conditions.

2. Discuss potential societal risks of multimodal sentiment analysis.
Systems that infer emotions or sentiment from multimodal signals can introduce risks such as:
a. Bias and fairness issues, particularly if models perform differently across demographic groups.
b. Privacy concerns, especially when analyzing audio or visual signals from individuals.
c. Misinterpretation of affective states, which could have consequences in sensitive applications (e.g., hiring, surveillance, or mental health monitoring).

3. Clarify safe and appropriate application contexts.
The authors should describe scenarios where the technology should be used cautiously and emphasize that predictions of emotional states should not be treated as definitive psychological assessments.

**Strengths And Weaknesses:**

Strengths
1. The paper is generally technically sound and methodologically reasonable. The proposed API framework is clearly defined and the algorithmic components that SCOPE (prototype estimation) and DIAM (instance-adaptive modulation) are mathematically specified with clear training objectives and procedures. The formulation of prototype construction, contrastive learning, and affine modulation is consistent with established techniques in multimodal learning and representation learning.

2. The experimental design is relatively comprehensive. The authors evaluate the method on two widely used benchmarks (CMU-MOSI and CMU-MOSEI), examine multiple missing-modality scenarios, and compare against several representative recovery-based and non-recovery-based baselines. Performance is evaluated across a range of missing rates (0.1–0.7), which provides insight into robustness under varying levels of incompleteness.

3. In addition to accuracy metrics, the authors provide ablation studies, qualitative visualizations, efficiency analysis, and hyperparameter sensitivity experiments. These analyses help support the claimed benefits of semantic anchoring and instance-adaptive modulation. The efficiency comparison (parameters, FLOPs, latency) is particularly useful for validating the claimed advantage over diffusion-based recovery methods.

Weaknesses
1. Limited evaluation scope. Experiments are restricted to only two datasets in a single task domain (multimodal sentiment analysis). While MOSI and MOSEI are standard benchmarks, the absence of additional datasets or tasks (e.g., emotion recognition, multimodal reasoning, or missing-modality video understanding) limits the generality of the conclusions.

2. Prototype construction assumptions. The method discretizes continuous sentiment labels into seven classes to enable prototype learning. This design choice introduces an approximation that may affect regression fidelity. The paper does not thoroughly analyze the effect of discretization or compare with alternative approaches (e.g., continuous prototype learning).

3. Limited theoretical justification. The paper lacks formal theoretical analysis explaining why prototype anchoring guarantees semantic consistency or why affine modulation is sufficient for recovering modality-specific features.

4. Potential training–inference mismatch. During training, the ground-truth class is used in the prototype selection step, while inference relies on predicted class similarity. The impact of this discrepancy is not carefully studied.

---

> ### Author Rebuttal · Authors · 2026-03-31
>
> Dear Reviewer LkTn
>
> Thank you for your thoughtful review and for recognizing our method's technical soundness and efficiency.
>
> > `Weakness 1 & Question 1`: Additional dataset.
>
> We additionally evaluate API on the emotion recognition benchmark IEMOCAP under the same random-missing protocol, where API consistently achieves better results than the state-of-the-art baseline HME across all missing rates.
>
> | mr | **API (ACC2/F1)** | HME (ACC2/F1) |
> | --- | ----------------------: | ------------: |
> | 0.1 | **85.2/85.2** | 80.9/80.3 |
> | 0.2 | **83.5/83.4** | 80.7/80.0 |
> | 0.3 | **82.9/82.8** | 79.0/77.8 |
> | 0.4 | **81.1/81.0** | 78.7/77.3 |
> | 0.5 | **79.5/79.5** | 78.5/77.1 |
> | 0.6 | **79.0/78.9** | 77.8/76.1 |
> | 0.7 | **77.9/78.0** | 77.6/76.0 |
>
> > `Weakness 2 & Question 2`: Discretization and continuous prototypes.
>
> Our choice of OSD follows the label structure of CMU-MOSI and CMU-MOSEI. Their sentiment annotations are natively based on a 7-level Likert-style scale over [-3, 3]. Thus, discretizing labels into seven ordinal classes preserves the original annotation granularity. Since the number of bins is tied to the dataset annotation rather than treated as a tunable hyperparameter, we did not include a separate sensitivity study over bin numbers.
>
> A continuous-prototype formulation is possible, but it is less aligned with our goal of building stable class-level semantic anchors. Continuous prototypes are learnable parameters mainly supervised by the main task, which makes them less suitable for capturing shared class priors. In contrast, our statistical prototypes are directly aggregated from samples within the same sentiment level, making them a more natural choice for modeling shared class-level semantics.
>
> > `Question 3`: Extreme two-missing scenario.
>
> When two modalities are missing and only one remains observed, API uses the sole observed modality to identify the most likely class via JMCS, and the corresponding direction-specific DIAM branches then impute the two missing modalities separately. For example, if only text is observed, API computes its similarity to the text prototype bank, selects the most likely class, and then uses the text->audio and text->vision branches to retrieve and modulate the corresponding audio and vision prototypes. We will clarify this process more explicitly in the final version.
>
> > `Weakness 3`: Limited theoretical justification.
>
> Below we make this statement precise.
>
> For modality $m$, let
>
> $$ p_j^m=\\frac{\\mu(P_j^m)}{\\|\\mu(P_j^m)\\|_2}. $$
>
> $$ z_i^m=\\frac{\\mu(\\tilde X_i^m)}{\\|\\mu(\\tilde X_i^m)\\|_2}. $$
>
> $$ \\Delta_c^m=\\min_{j\\ne c}\\|p_c^m-p_j^m\\|_2. $$
>
> If JMCS selects class $c$ and
>
> $$ \\|z_i^m-p_c^m\\|_2 < \\Delta_c^m/2, $$
>
> then for any $j\ne c$,
>
> $$ \\|z_i^m-p_j^m\\|_2 \\ge \\|p_c^m-p_j^m\\|_2-\\|z_i^m-p_c^m\\|_2 > \\Delta_c^m/2 > \\|z_i^m-p_c^m\\|_2. $$
>
> Since normalized descriptors satisfy $\\|u-v\\|_2^2=2-2\\cos(u,v)$, it follows that
>
> $$ c=\\arg\\max_j \\cos(z_i^m,p_j^m). $$
>
> Thus, under correct prototype retrieval and the half-margin condition, prototype anchoring preserves the selected prototype-based ordinal/class assignment in the normalized pooled-descriptor space.
>
> For a fixed selected prototype $P_c^t$, define
>
> $$ \\mathcal F_c^t=\\{\\Gamma\\odot P_c^t+B \\mid \\gamma,\\beta\\in\\mathbb R^{d_t}\\}. $$
>
> If $X_i^t\\in\\mathcal F_c^t$, affine modulation is exact. Otherwise, the best element in $\\mathcal F_c^t$ is the channel-wise least-squares projection
>
> $$ (\\gamma_k^\\star,\\beta_k^\\star)=\\arg\\min_{\\gamma_k,\\beta_k}\\|x_k-\\gamma_k p_k-\\beta_k\\mathbf 1\\|_2^2. $$
>
> Thus, for a fixed selected prototype, affine modulation is exact within the prototype-conditioned affine family and otherwise defines the best affine approximation target in that family, which $L_{\\text{rec}}$ trains DIAM to approximate.
>
> > `Weakness 4`: Training-inference mismatch.
>
> We would like to clarify that using the ground-truth class during training is intended to stabilize prototype-guided learning, rather than introducing a fundamentally different inference mechanism. In particular, SCOPE's prototype-contrastive objective explicitly shapes the same prototype-similarity space later used by JMCS at inference: each modality feature is pulled toward the prototype of its ground-truth class and pushed away from prototypes of other classes. Therefore, inference still relies on a similarity structure that has been directly optimized during training, rather than on an unrelated criterion. We will clarify this rationale more explicitly in the final version.
>
> Impact Statement.
>
> The revised Impact Statement is:
>
> "This work studies incomplete multimodal sentiment analysis under missing modalities. Risks include demographic bias, privacy concerns, and misinterpretation of affective states. Such systems should be used cautiously in high-stakes settings and not treated as definitive psychological assessments or sole decision criteria."

---

> > ### Author Rebuttal · Reviewer_LkTn · 2026-04-02
> >
> > The author's reply was very detailed and resolved most of my doubts.
> >
> > Update Score: Weak Accept

---

> > > ### Author Response · Authors · 2026-04-03
> > >
> > > We sincerely appreciate the reviewer’s recognition of our rebuttal. We would be grateful if you could kindly update the score at your convenience. Should you have any further questions or concerns, we are pleased to provide further clarification. Thank you.

---

### Official Review · Reviewer_wiPm · 2026-03-13

**Soundness:** 2
**Presentation:** 3
**Significance:** 2
**Originality:** 2
**Overall Recommendation:** 3
**Confidence:** 5

**Summary:**

This paper focuses on multimodal sentiment analysis with incomplete modalities. To overcome the limitations of both recovery-based and non-recovery approaches, the authors propose a framework called API. The framework first learns a class-specific prototype. When predicting with missing modalities, it retrieves the corresponding class prototype and adapts it using instance-level modulation. Experiments on the CMU-MOSI and CMU-MOSEI demonstrate the effectiveness of the proposed method, which also achieves lower latency compared to existing approaches.

**Compliance With Llm Reviewing Policy:**

Affirmed.

**Final Justification:**

The authors' response fails to adequately address my concerns. While the authors mention some works on MSA under missing modalities, the current research remains focused on mimic scenarios, which deviate significantly from real-world settings. Furthermore, the marginal improvements and lack of reproducibility lead me to recommend rejection. I do not believe this paper meets the high standards of ICML.

**Key Questions For Authors:**

Please see my comments on weaknesses.

**Limitations:**

The manuscript does not explicitly discuss any limitations of the proposed approach.

**Strengths And Weaknesses:**

Strengths:

1.	The paper introduces a framework that leverages prototype learning to handle missing modalities.

2.	The proposed method demonstrates low latency.

3.	The paper is well-written and clearly organized.

Weaknesses:

1.	The primary novelty of this paper lies in incorporating prototype learning, transforming the traditional one-stage estimation process into a two-stage approach. Specifically, it first estimating a class prototype, and then using instance-level modulation. However, extending the pipeline introduces potential drawbacks, such as error propagation across stages, which could negatively impact overall performance.

2.	As shown in Table 1, the proposed method offers only slight performance gains over existing baselines, suggesting limited practical effectiveness. Additionally, statistical significance tests are not reported, making it difficult to assess whether the observed improvements are meaningful.

3.	From my perspective, this paper lacks strong motivation. First, it claims that recovery-based methods suffer from a lack of class-level priors. While API initially estimates a class-level prototype, it may still rely on an incorrect class-level prototype during the DIAM step. As for non-recovery methods, the paper argues that they employ a static, global projection. In contrast, API addresses this issue by using an instance-specific DIAM. The key difference is replacing a single global projection with a two-step process: first estimating an instance-specific parameter, and then using that parameter to estimate the missing modality. However, in my view, this distinction is not different from what existing works have already done.

4.	In the experimental setup, the paper randomly masks one or two modalities. However, in real-world scenarios, such cases rarely occur. For example, audio and text data are often presented together. Therefore, the paper should discuss the performance gap between simulated and real-world conditions, and evaluate whether the proposed method remains effective in practical missing-modality situations.

---

> ### Author Rebuttal · Authors · 2026-03-31
>
> Dear Reviewer wiPm
>
> Thank you for your detailed review and constructive comments. We appreciate the opportunity to clarify these issues and provide additional evidence.
>
> > `Weakness 1 & 3`: Two-stage novelty and error propagation
>
> We would like to clarify that the novelty of our method is not simply the use of two stages, but the explicit decomposition of incomplete multimodal sentiment imputation into semantic anchoring and instance adaptation. Our core intuition is that affective expressions contain both class-shared commonality and sample-specific differences. Samples within the same sentiment class often share common affective patterns, while still exhibiting instance-specific differences in expression style and intensity. A one-stage or global mapping must infer these two factors simultaneously from incomplete observations, which tends to entangle class-level semantic consistency with sample-specific expression differences. API is therefore a retrieval-and-modulation formulation rather than a superficial pipeline extension.
>
> On the recovery-based side, SCOPE constructs class prototypes as statistically derived representations aggregated over many training samples, serving as stable class-level semantic anchors. Moreover, SCOPE uses a prototype-contrastive objective that pulls each modality feature toward the prototype of its ground-truth class while pushing it away from prototypes of other classes in the same modality, which makes selecting an incorrect class prototype less likely. Therefore, the two-stage design does not naturally lead to strong error propagation; DIAM is a lightweight refinement constrained by reconstruction supervision, not an unconstrained generator. This is also consistent with Table 2: if the second stage mainly introduced cascading errors, adding DIAM on top of SCOPE should hurt performance, but instead it consistently improves over SCOPE on both MOSI and MOSEI.
>
> On the non-recovery-based side, API is not replacing a global projection with one shared instance-specific module. DIAM uses six direction-specific source-to-target groups, where each group first projects the source cue into the target-modality space and then generates affine parameters to modulate the retrieved class prototype. During training, DIAM is supervised to learn how class-consistent prototypes should be adapted to individual samples across different cross-modal directions, and this training design makes it easier for DIAM to bridge class-shared affective patterns and sample-specific expression differences. Therefore, API is materially different from prior methods that rely on a single static global projection.
>
> We will revise the paper to make this motivation and its distinction from prior one-stage methods more explicit.
>
> > `Weakness 2`: Practical effectiveness.
>
> While the gains in Table 1 may appear modest in isolation, API still attains the best average results on MOSI/MOSEI and offers large advantages in learnable parameters and latency (Table 2; Sec. 4.4).
>
> To further substantiate practical effectiveness, we further evaluate API on the widely used emotion-recognition benchmark **IEMOCAP**, with results averaged over five random seeds, where API consistently achieves better results than the state-of-the-art baseline HME in all fixed- and random-missing settings.
>
> Fixed missing:
>
> | Avail | **API (ACC2/F1)** | HME (ACC2/F1) |
> | --- | ---: | ---: |
> | L | **83.9/84.0** | 79.1/78.1 |
> | V | **77.2/77.1** | 75.7/70.8 |
> | A | **80.0/79.6** | 78.9/76.8 |
> | L,V | **86.1/86.1** | 79.7/78.9 |
> | L,A | **86.4/86.4** | 80.7/80.0 |
> | A,V | **80.2/79.9** | 78.1/76.3 |
>
> Random missing:
>
> | mr | **API (ACC2/F1)** | HME (ACC2/F1) |
> | --- | ---: | ---: |
> | 0.1 | **85.2/85.2** | 80.9/80.3 |
> | 0.2 | **83.5/83.4** | 80.7/80.0 |
> | 0.3 | **82.9/82.8** | 79.0/77.8 |
> | 0.4 | **81.1/81.0** | 78.7/77.3 |
> | 0.5 | **79.5/79.5** | 78.5/77.1 |
> | 0.6 | **79.0/78.9** | 77.8/76.1 |
> | 0.7 | **77.9/78.0** | 77.6/76.0 |
>
> > `Weakness 4`: Practical missing-modality settings.
>
> We follow the random-missing protocol for fair comparison with prior work; to address structured real-world missingness, we additionally evaluate a **fixed-missing protocol** on both MOSI and MOSEI, where API consistently performs better than HME.
>
> MOSI:
>
> | Avail | **API (ACC2/F1)** | HME (ACC2/F1) |
> | --- | ---: | ---: |
> | L | **85.8/85.8** | 83.9/83.8 |
> | V | **64.5/64.4** | 57.8/54.6 |
> | A | **63.3/63.2** | 57.8/57.7 |
> | L,V | **86.1/86.0** | 84.3/84.3 |
> | L,A | **86.3/86.1** | 84.4/84.4 |
> | A,V | **65.2/65.2** | 57.8/55.5 |
>
> MOSEI:
>
> | Avail | **API (ACC2/F1)** | HME (ACC2/F1) |
> | --- | ---: | ---: |
> | L | **85.5/85.4** | 84.7/84.7 |
> | V | **66.7/65.4** | 62.9/53.0 |
> | A | **65.1/65.5** | 63.0/50.4 |
> | L,V | **85.6/85.6** | 84.8/84.8 |
> | L,A | **85.7/85.5** | 84.4/84.4 |
> | A,V | **66.1/65.0** | 62.9/55.6 |

---

> > ### Author Rebuttal · Reviewer_wiPm · 2026-04-04
> >
> > Thanks for your response. However, my concerns remain unresolved.
> >
> > I acknowledge that current research in incomplete multimodal emotion recognition typically relies on random missing. However, as other reviewers have noted, a significant gap exists between these simulated conditions and real-world scenarios. This highlights the domain limitations, leading me to question whether we should continue conducting efforts in this specific direction.
> >
> > Meanwhile, from my perspective, the paper still lacks a strong underlying motivation. Experimental results in Table 2 show only marginal gains, even if the API performs well elsewhere. Meanwhile, the absence of source code prevents reproduction; this is a common issue in the field but further undermines confidence in the reliability of your experiments.
> >
> > Therefore, I still do not believe this paper meets the high standard of ICML.

---

> > > ### Author Response · Authors · 2026-04-04
> > >
> > > Dear Reviewer wiPm,
> > >
> > > Thank you for your follow-up and for the careful consideration you have given to our paper. We sincerely appreciate your time and thoughtful feedback.
> > >
> > > > `Follow-up concern`: the value of studying incomplete MSA under "these simulated conditions"
> > >
> > > We agree that these simulated conditions do not fully cover all unconstrained real-world missingness. Nevertheless, similar settings have been widely adopted in prior work on incomplete multimodal sentiment analysis. The recent methods already cited in our paper, including CorrKD (CVPR 2024), LNLN (NeurIPS 2024), HME (NeurIPS 2025), and M2AF (SIGIR 2025), are all evaluated under the same general simulated missing setting. Our paper therefore follows the evaluation protocol commonly used in prior work rather than relying on a setting specific to our method.
> > >
> > > To connect these simulated conditions more closely to real-world scenarios, we further added fixed-missing experiments in addition to the standard random-missing evaluation. Taken together, these two settings provide a closer approximation to common real-world missing patterns while keeping comparisons aligned with prior work. More importantly, even under these standard simulated settings, there are still substantial open problems in incomplete MSA that existing methods have not resolved well, and we believe it remains important to continue making progress in this direction.
> > >
> > > > `Practical effectiveness`
> > >
> > > We understand the concern that the gains over prior methods may appear moderate in some settings. We would like to emphasize that API achieves better results on the standard benchmarks CMU-MOSI and CMU-MOSEI, and our additional IEMOCAP results further support the robustness of API beyond these two datasets. In addition, API maintains a clear efficiency advantage: SCOPE introduces no learnable parameters, DIAM is lightweight, and the missing-modality module requires only about 0.14M / 0.05M learnable parameters on MOSI / MOSEI while maintaining very low latency. We believe this combination of improved performance and strong efficiency is the main practical value of the paper.
> > >
> > > To further improve reproducibility, we will release code.

---

### Official Review · Reviewer_Xok1 · 2026-03-16

**Soundness:** 4
**Presentation:** 4
**Significance:** 4
**Originality:** 4
**Overall Recommendation:** 5
**Confidence:** 5

**Summary:**

The paper proposes the Adaptive Prototype Imputation (API) framework to deal with missing modalities in multimodal learning. The paper observes the current limitations of recovery-based methods (computational inefficiency and semantic inconsistency) and non-recovery-based methods (lack of instance specificity). API aims to address with two key components: Semantic-anchored Class-Temporal Prototype Estimation (SCOPE), and Directional Instance-Adaptive Affine Modulation (DIAM). The extension of prototypes to the regression task is an interesting contribution.

**Compliance With Llm Reviewing Policy:**

Affirmed.

**Final Justification:**

This is a good paper.

**Key Questions For Authors:**

It is not clear what the train/test partition is. Can you clarify?

**Limitations:**

not addressed.

**Strengths And Weaknesses:**

Strenghs

The explanation of the methods starts with a clear motivation before detailing the methods. This makes the paper easy to follow.

The evaluation is very exhaustive.


Major limitations

I have a comment about the assumption of missing frames. The frames are not randomly missed in practice. Participants turn their faces, and their faces are gone for a few consecutive frames. A noise appears in the audio and persists over several frames. Therefore, a more realistic scenario is that if a frame is missing, the chance of the next frame being missed increases. This modeling is more reasonable and realistic. Also, it is common for all modalities to be lost over a few frames (a package on the internet is lost). Therefore, the assumption that you always have 1 modality is not realistic (“We strictly enforce m_i >=1”).

The paper says, “it lacks instance-level granularity. DIAM bridges this gap by applying dynamic affine transformations to these anchors” (Section 4.2). What are the results that you are using to conclude this claim? This is an important goal of your model, but I am not convinced that the results in the analysis support this feature of the model. More specifically, it is not clear to me how Figure 6 demonstrates that DIAM generates instance-specific modulations rather than a static bias.

For the section “Distributional Fidelity of Imputation.” It would be useful to provide numerical support for the qualitative descriptions in Figure 4. For example, you can report the correlation between the ground truth and prototypes.


It is not clear what the train/test partition is. Can you clarify?

---

> ### Author Rebuttal · Authors · 2026-03-31
>
> Dear Reviewer Xok1
>
> We are deeply grateful for your positive feedback on our work and the insightful suggestions. We have carefully reviewed each point and provided detailed responses accordingly.
>
> > `Question 1`: Missing-Modality Setting and Scope
>
> We fully agree that temporally correlated missingness, such as consecutive missing frames, is realistic and important. We would like to clarify that two commonly studied missing-modality settings are inter-modality missingness and intra-modality missingness. Our paper focuses on the former, where one or more entire modalities of an utterance are unavailable while at least one modality remains observed. This is the standard setting adopted in prior incomplete-MSA work such as IMDer and HME. By contrast, consecutive missing frames, or short spans where all modalities are lost, are closer to frame-level/intra-modality missingness and are beyond the scope of the current paper.
>
> Inter-modality missingness is still practically meaningful. In real applications, an entire modality of an utterance may be unavailable due to camera-off, ASR transcript failure, or microphone corruption.
>
> Regarding "we strictly enforce m_i >= 1", we agree that this assumption should be stated more clearly. If all modalities of an utterance are absent, there is no valid information for the input in that sample. We will revise the paper to make the target setting and this scope boundary clearer.
>
> > `Question 2`: Explanation of Instance-Adaptive Modulation
>
> The empirical support for this claim comes from both the ablation in Sec. 4.2/Table 2 and the qualitative analysis in Sec. 4.3/Figure 4. In Table 2, SCOPE already provides class-shared semantic anchors, and adding DIAM consistently yields further improvements on both datasets (MOSI F1 76.9 -> 79.6; MOSEI F1 74.0 -> 77.8). Figure 4 provides complementary evidence by showing the distribution alignment between imputed instance-adaptive prototypes and the corresponding ground-truth features, supporting our claim that DIAM bridges this gap by applying dynamic affine transformations to these anchors.
>
> This interpretation also matches the design of DIAM in Sec. 3.2. DIAM uses six direction-specific source-to-target groups, where each group projects the observed source cue into the target-modality space and then generates affine parameters to modulate the retrieved class prototype. During training, DIAM is supervised to learn how class-consistent prototypes should be adapted to individual samples across cross-modal directions, making it easier to bridge class-shared affective patterns and sample-specific expression differences. Importantly, the scaling and shifting parameters ($\gamma, \beta$) are not global fixed parameters; they are generated from the current sample's observed source cue through DSP/APG, so the modulation is conditioned on the current input rather than a single static bias shared across all samples. Figure 6 is intended as a qualitative illustration of this mechanism: it visualizes the generated scaling/shifting parameters under two different source-target directions (Vision -> Text and Audio -> Vision). We will revise Sec. 4.2 to make this evidence chain clearer.
>
> > `Question 3`: Quantitative Support for Distributional Fidelity
>
> We agree that quantitative support would strengthen the "Distributional Fidelity of Imputation" analysis. As suggested, we additionally compute quantitative correlations for this analysis. Specifically, we randomly sampled 1,500 examples from the CMU-MOSEI test set, temporally averaged both the ground-truth feature and the corresponding instance-adaptive prototype for each sample, computed the Pearson correlation between the two vectors, and then reported the mean correlation over samples. The resulting mean correlations are **0.857** for text, **0.856** for audio, and **0.664** for vision. These results provide quantitative support for the qualitative overlap in Figure 4 and show that the imputed instance-adaptive prototypes remain substantially aligned with the corresponding ground-truth features in the original feature space.
>
> > `Question 4`: Train/Validation/Test Partition
>
> We indeed omitted this detail in the current draft. We use the official default train/validation/test split provided by the CMU Multimodal SDK, following prior work. Specifically, CMU-MOSI uses 1,284/229/686 samples for train/validation/test, and CMU-MOSEI uses 16,326/1,871/4,659. We train on the training split, select checkpoints on the validation split, and report final results on the test split. We will add this clarification to Sec. 4.
>
> We thank you again for your constructive feedback, which has helped us improve the clarity and completeness of the paper.

---

> > ### Author Rebuttal · Reviewer_Xok1 · 2026-04-03
> >
> > Thanks for the clarification.

---

> > > ### Author Response · Authors · 2026-04-04
> > >
> > > Dear Reviewer Xok1,
> > >
> > > Thank you for your acknowledgement and for confirming that our clarifications resolved your concerns. We will make the relevant clarifications explicit in the final version to further improve clarity and completeness.

---

### Decision · Program_Chairs · 2026-04-30

**Decision:**

Accept (regular)

**Comment:**

This paper proposes the Adaptive Prototype Imputation (API) framework to deal with missing modalities in multimodal sentiment analysis. This paper proposes Semantic-anchored Class-Temporal Prototype Estimation (SCOPE), and Directional Instance-Adaptive Affine Modulation (DIAM), to deal with the limitations of recovery-based methods (computational inefficiency and semantic inconsistency) and non-recovery-based methods (lack of instance specificity), respectively. Four reviewers reviewed this paper with the overall ratings of one accept, two weak accept, and one weak reject. The reviewers recognized the low latency, good organization and presentation, technically sound method, comprehensive experimental design, and sufficient experiments. Meanwhile, they pointed out some critical concerns about different aspects of this paper, such as the impractical assumption of missing frames, unclear or undefined notations, ambiguous description of the cross-dataset generalization experiment, marginal performance gains, limited evaluation scope, prototype construction assumptions, limited theoretical justification, and potential training–inference mismatch. The rebuttal properly addressed most of the concerns, and reviewers Xok1, LkTn, jMTd are generally satisfied with the rebuttal. Reviewer wiPm still has some major concerns on the impractical real-world settings, marginal improvements, and lack of reproducibility. The AC thinks this paper makes good progress on the balance of efficiency and effectiveness for incomplete multimodal sentiment analysis. Please take the reviewers' comments and suggestions into consideration when preparing the final version.